# Chemical Constituents of *Macaranga occidentalis*, Antimicrobial and Chemophenetic Studies

**DOI:** 10.3390/molecules27248820

**Published:** 2022-12-12

**Authors:** Viviane Flore Kamlo Kamso, Christophe Colombe Simo Fotso, Ines Michèle Kanko Mbekou, Billy Tchegnitegni Tousssie, Bruno Ndjakou Lenta, Fabrice Fekam Boyom, Norbert Sewald, Marcel Frese, Bonaventure Tchaleu Ngadjui, Ghislain Wabo Fotso

**Affiliations:** 1Department of Organic Chemistry, Faculty of Science, University of Yaoundé I, Yaoundé P.O. Box 812, Cameroon; 2Department of Chemistry, Faculty of Science, University of Buea, Buea P.O. Box 63, Cameroon; 3Department of Biochemistry, Faculty of Science, University of Yaoundé I, Yaoundé P.O. Box 812, Cameroon; 4Department of Chemistry, Faculty of Science, University of Dschang, Dschang P.O. Box 67, Cameroon; 5Department of Chemistry, Higher Teacher Training College, University of Yaoundé I, Yaoundé P.O. Box 47, Cameroon; 6Department of Chemistry, Organic and Bioorganic Chemistry, University of Bielefeld, 33501 Bielefeld, Germany

**Keywords:** *Macaranga occidentalis*, prenylated flavonoids, stilbenes, antimicrobial activity, pharmacomodulation, chemophenetic significance

## Abstract

Medicinal plants are known as sources of potential antimicrobial compounds belonging to different classes. The aim of the present work was to evaluate the antimicrobial potential of the crude extract, fractions, and some isolated secondary metabolites from the leaves of *Macaranga occidentalis*, a Cameroonian medicinal plant traditionally used for the treatment of microbial infections. Repeated column chromatography of the ethyl acetate and n-butanol fractions led to the isolation of seventeen previously known compounds (**1−17**), among which three steroids (**1−3**), one triterpene (**4**), four flavonoids (**5−8**), two stilbenoids (**9** and **10**) four ellagic acid derivatives (**11−14**), one geraniinic acid derivative (**15**), one coumarine (**16**), and one glyceride (**17**). Their structures were elucidated mainly by means of extensive spectroscopic and spectrometric (1D and 2D NMR and, MS) analysis and comparison with the published data. The crude extract, fractions, and isolated compounds were all screened for their antimicrobial activity. None of the natural compounds was active against *Candida* strains. However, the crude extract, fractions, and compounds showed varying levels of antibacterial properties against at least one of the tested bacterial strains, with minimal inhibitory concentrations (MICs) ranging from 250 to 1000 μg/mL. The *n*-butanol (*n*-BuOH) fraction was the most active against *Escherichia coli* ATCC 25922, with an MIC value of 250 μg/mL. Among the isolated compounds, schweinfurthin B (**10**) exhibited the best activity against *Staphylococcus aureus* NR 46003 with a MIC value of 62.5 μg/mL. In addition, schweinfurthin O (**9**) and isomacarangin (**6**) also exhibited moderate activity against the same strain with a MIC value of 125 μg/mL. Therefore, pharmacomodulation was performed on compound **6** and three new semisynthetic derivatives (**6a–c**) were prepared by allylation and acetylation reactions and screened for their in vitro antimicrobial activity. None of the semisynthetic derivatives showed antimicrobial activity against the same tested strains. The chemophenetic significance of the isolated compounds is also discussed in this paper.

## 1. Introduction

Infectious diseases caused by bacteria, viruses, fungi, and other parasites continue to cause enormous damage worldwide. Bacterial infections kill over seven million people annually, and they may kill up to 10 million people by the year 2050 if appropriate measures are not taken [1]. The high rate of bacterial resistance to available antibiotics is alarming and makes the treatment of even simple bacterial infections difficult [2]. This resistance could be due to the capacity of Gram-positive or Gram-negative bacteria to acquire resistance mechanisms to face environmental aggression such as competing bacteria, the natural environment, host defense, or antibiotics, either by modification of the anti-infective active sites or by the production of degradative enzymes [3]. This concern requires a continuous search for new and efficient lead antibacterial agents to fight against multiresistant microbial agents and to limit undesirable side effects. Plants have long been reported as important sources of bioactive molecules [4]. Plants of the *Macaranga* genus of the Euphorbiaceae are commonly used by traditional healers for the treatment of various diseases such as swellings, cuts, sores, diarrhea, cough, stomach-ache, hypertension, boils, furuncles, and bruises [5,6,7,8]. *Macaranga occidentalis* (Müll.Arg.) Müll.Arg. is used in the western region of Cameroon to treat stomach wash for pregnant women. Previous pharmacological studies of the crude extracts, fractions, and isolated compounds of the *Macaranga* genus possess a wide range of biological activities including anticancer, antioxidant, anti-inflammatory, and antimicrobial activities [5]. Chemical investigations of plants of this genus indicate that they constitute a rich source of isoprenylated, geranylated, and farnesylated flavonoids and stilbenes [7,8], terpenoids [9], coumarins [10], ellagic acid derivatives, and tannins [11]. As part of our long-term research work on bioactive natural product medicinal plants [12,13], we examined the leaves of *M. occidentalis* growing in Cameroon. We herein report the antimicrobial potential of the crude extract, ethyl acetate (EtOAc), and *n*-BuOH fractions of this plant from which seventeen naturally occurring compounds were isolated and screened. In addition, three new semisynthetic derivatives were prepared and screened for their in vitro antimicrobial potential. To the best of our knowledge, this work also provides the first chemical and biological investigation of *M. occidentalis* as well as the chemophenetic significance of the isolated compounds.

## 2. Results and Discussion

### 2.1. Isolation of Specialized Metabolites from M. occidentalis

The DCM–MeOH (1:1, *v/v*) extract from the leaves of *M. occidentalis* was subjected to liquid–liquid partition with EtOAc and *n*-BuOH. Repeated column chromatography on silica gel and Sephadex LH-20 of these fractions afforded 17 known metabolites (**1**–**17**) (Figure 1). Their structures were established by spectroscopic (1D and 2D NMR) and spectrometric analysis and by comparison with the literature data as a mixture of *β*-sitosterol and stigmasterol (**1** and **2**) [14], *β*-sitosterol-3-*O*-*β*-*D*-glucopyranoside (**3**) [15], lupeol (**4**) [16], apigenin-7-*O*-*β*-*D*-glucopyranoside (**5**) [17], isomacarangin (**6**) [6], kaempferol (**7**) [18], quercetin (**8**) [18]; schweinfurthin O (**9**) [8], schweinfurthin B (**10**) [19], ellagic acid (**11**) [11], 3,4-methylenedioxy-3′-*O*-methylellagic (**12**) [20], 3,3′,4-tri-*O*-methylellagic acid 4′-*O*-*β*-D-glucopyranoside (**13**) [21], 3,3′,4′-tri-*O*-methylellagic acid (**14**) [22], (5*R*,6*R*)-4,6-dihydrocarbonyl-5-[2′,3′,4′-trihydroxy-6′-(methoxycarbonyl)phenyl]-5,6-dihydro-2H-pyran-2-one (**15**) [23], methyl brocchllin carboxylate (**16**) [24], and ishigoside (**17**) [25].

### 2.2. Antimicrobial Activity of the Extract, Fractions, and Isolated Compounds

The results of the in vitro antibacterial activities of the DCM-MeOH (1:1, *v/v*) extract, EtOAc and *n*-BuOH fractions as well as the isolates and semisynthetic compounds are presented in Table 1. All extracts and fractions showed varying levels of antibacterial properties against at least one of the tested bacterial strains, with MICs ranging from 250 to 1000 μg/mL. The antibacterial activity of the plant extract and fraction can be classified as significant (MIC <100 μg/mL), moderate (100 < MIC < 625 μg/mL) and weak (MIC > 625 μg/mL) [26]. According to this classification, the inhibitory potential of the screened extract and fractions could be considered moderate to weak. The *n*-BuOH (MIC = 250 μg/mL) fraction was the most active against *Escherichia coli* ATCC 25922, followed by the MeOH extract (MIC = 500–1000 μg/mL), which displayed activity against *Staphylococcus aureus* NR 46003, *Escherichia coli* ATCC 25922, and *Shigella flexneri* NR 518. Additionally, the EtOAc fraction also showed moderate activity (MIC = 500 μg/mL) against *Escherichia coli* ATCC 25922. The results obtained present the differences between the antibacterial activities of the extract and fractions from the *M. occidentalis* leaves. This suggests that the plant contains several active principles with different polarities, as shown by the nature of the isolates. Indeed, the antibacterial activity of medicinal plants is correlated with their chemical composition [3]. The *n*-BuOH fraction was the most active against *Escherichia coli* ATCC 25922, but all of the isolated compounds from this fraction presented weak or no activity against the selected strains, which could be due to the synergetic effect of the different constituents of this fraction. However, the purification of the EtOAc fraction afforded more active compounds (**6**, **8**, **9**, and **10**) against *Staphylococcus aureus* NR 46003 and *Shigella flexneri* NR 518 and suggested the antagonist effect of compounds in this fraction. EtOAc is a semipolar solvent and can effectively extract semipolar active compounds such as flavonoids, stilbenes, terpenoids, and ellagic acids, which are well-known to have a broad spectrum of activity against bacterial strains [27]. The results obtained herein are in agreement with those reported in the literature [28,29], which present the antimicrobial potential of the methanolic extracts of *M. gigantea*, *M. pruinosa*, *M. tanarius*, and *M. triloba* using the disc-diffusion method against Gram-positive bacteria (*Bacillus cereus*, *Micrococcus luteus* and, *Staphylococcus aureus*) and Gram-negative bacteria (*Escherichia coli*, *Klebsiella pneumoniae* and, *Salmonella choleraius*). In this study, all the *Macaranga* extracts exhibited moderate inhibition diameters and only for Gram-positive species. Similar results were obtained by Salah et al. (2003) after screening of the Cameroonian *M. monandra*, which showed that it was also inactive against *C. acutatum*, *C. gloeosporioides,* and *C. fragariea* [30]. Our results not only validate the use of *M. occidentalis* in folk medicine to treat related diseases, but also support previous literature results from plants of this genus. Regarding the pure compounds, their antimicrobial activity can be classified as significant (MIC <10 μg/mL), moderate (10 < MIC < 100 μg/mL), or weak (MIC > 100 μg/mL) [26]. According to this point, the isolates showed inhibition ranging from moderate to weak. Schweinfurthin B (**10**), which exhibited moderate activity against *Staphylococcus aureus* NR 46003 with a MIC value of 62.5 μg/mL, was the most active. Furthermore, schweinfurthin O (**9**) and isomacarangin (**6**) similar to schweinfurthin B (**10**), which belong to the C-prenylated phenolic compounds, exhibited moderate activity against the same strain with a MIC value of 125 μg/mL. These results are in agreement with those in the literature, which indicated that the presence of C-prenyl groups in flavonoids and other phenolic compounds played an important effect on the inhibitory activity against bacterial strains [31]. In addition, the good activity of compound **6** (MIC = 125 μg/mL) compared to those of **7** and **8**, which also belong to flavonoids, is not surprising, since it has been reported that C-prenylated flavonoids are more hydrophobic than common flavonoids, facilitating the ability to penetrate the cell membrane, thus improving their action at the active site [32].

### 2.3. Alkylation and Acylation of Compound ***6***: Semisynthesis of Alkylated, and Acylated Derivatives ***6a–c***

Lipophilicity is an important factor for the absorption of a drug candidate through the membrane of a microbe. Lipophilic groups such as allyl or prenyl groups have been demonstrated to enhance access and affinity or inhibit RAS transduction [33,34,35]. Therefore, compound **6**, showing moderate antimicrobial activity, was allylated under weakly basic conditions, yielding two allylated derivatives **6a** and **6b,** as shown in Figure 1. The formation of compound **6b** can be explained by the Claisen rearrangement of the 7-O-allylphenyl ether moiety in **6a** to an O-allylphenol possessing a C-allyl substituent, as shown in Figure 2. To demonstrate the importance of the presence of phenolic groups for antimicrobial activity, compound **6** was acetylated using acetic anhydride in pyridine at room temperature for 24 h to afford tetraacetylated product **6c** (Figure 1).

#### 2.3.1. Characterization of Compounds **6a–c**

Compound **6a** was obtained as a yellowish finely divided solid soluble in acetone. Its molecular formula was determined to be C_34_H_38_O_6_, with 16 degrees of unsaturation, based on its NMR data and its HRESIMS data, which showed the sodium adduct peak [M + Na]^+^ at m/z 565.2560 (calcd for C_34_H_38_O_6_Na^+^: 565.2560). The IR spectrum showed characteristic absorption bands at 3461, 1739.8, 1650, and 1216.5 cm^−1^, indicating the presence of hydroxyl, carbonyl, olefinic double bonds, and ether groups, respectively [35]. The ^1^H-NMR spectrum (Table 2) showed the presence of an AA′BB′ system with four protons at δ_H_ 8.18 (H-2′/H-6′, J = 9.0 Hz) and 7.12 (d, H-3′/H-5′, J = 9.0 Hz); one aromatic singlet at δ_H_ 6.74 (s, H-6); and a chelated hydroxyl group at 12.87 ppm (OH-5). This spectrum also revealed the presence of three vinylic methyls as singlets at δ_H_ 1.55 (s, H-8″), 1.60 (s, H-9″), and 1.79 (s, H-10″); three vinylic methylenes at δ_H_ 3.38 (d, J = 7.2 Hz, H-1″), 1.96 (dd, J = 9.1, 6.3 Hz, H-4″), and 2.05 (m, H-5″) together with two olefinic protons at δ_H_ 5.26 (d, J = 1.5 Hz, H-2″) and 5.07 (m, H-6″). These signals were attributable to those of a geranyl group [35]. This was confirmed by HMBC correlations between H-10″ (1.79 ppm) and C-3″ (122.9 ppm), C-4″ (135.5 ppm) and C-5″ (40.2 ppm) on one hand, and between H-8″ and H-9″ and the similar carbons C-6″ (125.1 ppm) and C-7″ (131.6 ppm) on the other (Figure 2). The ^13^C NMR spectrum displayed characteristic signals of a kaempferol derivative at δ_C_ 138.2 (C-3), 156.0 (C-5), 131.1 (C-2′/C-6′), 115.4 (C-3′/C-5′), and 163.0 (C-7) [36,37]. The HMBC correlation from H-6 (6.74 ppm) and the aromatic carbons at 156.0 (C-5), 163.0 (C-7) and 113.0 (C-8) allowed us to suggest that the geranyl was located at position 8. This was further confirmed by the HMBC correlation between proton H-2″ (3.38 ppm) and carbons at 113.0 (C-8), 163.0 (C-7), and 158.6 (C-8a). All of these data were superimposable on those of isomacarangin (**6**) [36]. Careful examination of the remaining signals of ^1^H and ^13^C-NMR spectra together with ^1^H−^1^H COSY allowed us to identify three allyl groups by the characteristic signals of terminal olefinic methylenes at 118.2/5.30, 117.8/5.50, 117.9/5.52; olefinic methines at 134.8/6.00, 133.8/5.16, 134.2/6.13; and oxymethylene signals at δ_H/C_ 73.7/4.55, 70.0/4.75, and 69.5/4.69. The linkage of the allyl groups was evidenced by the HMBC correlations depicted from protons at 4.55, 4.75, 4.69 to carbons at 138.2 (C-3), 163.0 (C-7), and 161.6 (C-4′), respectively. According to these spectral data, the structure of **6a** was unambiguously elucidated as 3,7,4′-triallylisomacarangin.

Compound **6b** was also obtained as a yellowish finely divided solid soluble in acetone. Its molecular formula was determined to be C_37_H_42_O_6_, with 17 double bond equivalents, on the basis of its NMR data and its HRESIMS data, which showed the protonated adduct peak [M + Na]^+^ at m/z 583.3052 (calcd m/z 583.3054 for C_37_H_43_O_6_^+^). The ^1^H NMR spectrum of **6b** was closely related to that of **6a** (see Table 2) with the exception of the disappearance of the aromatic signal at 6.74 ppm on the A ring in **6a**, indicating a substitution at this position. Extensive analysis of the proton spectrum showed the presence of additional signals at δ_H_ 3.64, 6.05, and 7.14 ppm, which correlated in the HSQC with carbons at δ_C_ 28.5, 137.3, and 115.7 ppm, respectively, and indicated the presence of one C-allyl group in **6b**, which was further confirmed by the combined DEPT 135 and ^1^H−^1^H COSY (Figure 2) spectra. The linkage of the additional allyl group was evidenced by the HMBC correlation between the proton at δ_H_ 3.54 ppm and C-6 (112.3 ppm), C-5 (157.3) and C-7 (162.1). Consequently, compound **6b** was elucidated as 3,6,7,4′-tetraallylisomacarangin.

Compound **6c** was isolated as a yellowish finely divided solid soluble in acetone. Its NMR data in association with its HRESIMS showed the sodium adduct peak [M + Na]^+^ at m/z 613.2043 (calcd for C_33_H_34_O_10_Na^+^, 613.2044), which led to the assignment of C_33_H_34_O_10_, with 17 double bond equivalents, as the molecular formula of **6c**. The ^1^H-NMR spectrum (Table 1) of **6c** was closely related to those of isomacarangin (**6**) [35,36] with the signals of the AA′BB′ system of four protons at δ_H_ 8.81 (H-2′/H-6′, J = 8.8 Hz) and 8.16 (d, H-3′/H-5′, J = 8.8 Hz) and one aromatic singlet at δ_H_ 8.31 (s, H-6) and the characteristic signals of the geranyl group [35]. Extensive analysis of its ^1^H-NMR spectrum displayed the signals of four methyl protons at δ_H_ 2.31, 2.32, 2.39, and 2.40 ppm, which correlated in the HSQC spectrum with carbons at δ_C_ 20.4, 21.1, 20.4, and 21.0 ppm. The ^13^C-NMR spectrum of **6c** displayed four additional carbonyl signals at δ_C_ 169.4, 168.3, 169.2, and 168.4 ppm, indicating the presence of four acetyl units located in the four previously hydroxylated aromatic positions of isomacarangin (**6**). Therefore, compound **6c** was elucidated as 3,5,7,4′-tetraacetylisomacarangin.

#### 2.3.2. Antimicrobial Activity of Compounds **6a–c**

The semisynthetic derivatives obtained (**6a–c**) were screened against all the selected strains, and the results obtained reveal that all the semisynthetic compounds were less active (MIC >500 μg/mL) than isomacarangin (**6**) (MIC = 500 μg/mL). These results further confirmed the effect of the hydroxyl groups of flavonoids on the antibacterial activity against bacterial growth (Table 2).

### 2.4. Chemical Significance of the Isolated Compounds

The present study reports the first chemical investigation of *M. occidentalis*. To the best of our knowledge, this is the first report on the isolation of the specialized metabolites of *M. occidentalis*. However, some of the isolated compounds were reported from other species of the studied genus. This is the case for *β*-sitosterol (**1**), stigmasterol (**2**), and *β*-sitosterol-3-*O*-*β*-*D*-glucopyranoside (**3**) isolated from *M. magna* [14]; lupeol (**4**), evidenced from *M. balansae* [16]; schweinfurthins O and B (**9** and **10**), previously found in *M. tanarius,* and *M. schweinfurthii* [8,19]. Isomacarangin (**6**) was found in *M. barteri* [6] and *M. schweinfurthii* [36]; kaempferol (**7**) and quercetin (**8**) were isolated from *M. indica* [18]; ellagic acid (**11**) was found in *M. barteri* [11]. Nevertheless, this work reports, for the first time, the isolation of the glycosylated flavonoid apigenin-7-*O*-*β*-*D*-glycoside (**5**) from the *Macaranga* genus. However, this compound has already been isolated from other genera of the Euphorbiaceae family including *Euphorbia humifusa* [17] and *Chrozophora rottleri* [38]. Among the ellagic acid derivatives isolated from this plant, 3,4-methylenedioxy-3′-*O*-methylellagic (**12**), 3,3′,4-tri-*O*-methylellagic acid 4′-*O*-*β*-D-glucopyranoside (**13**), and 3,3′,4′-tri-*O*-methylellagic acid (**14**) were isolated for the first time in the Euphorbiaceae family. Furthermore, the presence of ellagic acid derivatives in the *Maracanga* genus was already reported by Mgoumfo et al. (2008) [11]. This work also reports the first isolation of the geraniinic acid derivative, (5*R*,6*R*)-4,6-dihydrocarbonyl-5-[2′,3′,4′-trihydroxy-6′-(methoxycarbonyl)phenyl]-5,6-dihydro-2H-pyran-2-one (**15**), and coumarin methyl brocchllin carboxylate (**16**) from the *Macaranga* genus. Furthermore, these two secondary metabolites have been isolated from *Phyllanthus reticulatus* and *Chrozophora brocchiana*, respectively, which belong to the Euphorbiaceae [23,24]. Nevertheless, the isolation of a coumarin from this plant is not surprising since two coumarins have previously been isolated from many *Macaranga* species such as *M. barteri* [11], *M. gigantifolia* [10], and *M. triloba* [39]. In addition, the isolated coumarin is a structural analog of the methyl brevifolin carboxylate previously isolated from *M. tanarius* [40]. This evidence suggests that there is a relationship between the genera *Macaranga*, *Phyllanthus*, and *Chrozophora* within the Euphorbiaceae family. We herein present the first isolation of ishigoside (**17**) in the Euphorbiaceae. This compound, which was isolated for the first time from the brown alga *Ishige okamurae* (Ishigeaceae) [25], was recently isolated from the terrestrial plant *Dracaena stedneuri* (Dracaenaceae) [41], thus, its presence in *M. occidentalis* is not surprising. Being cognizant of the fact that prenylated flavonoids and stilbenes constitute the chemotaxonomic markers of the *Macaranga* genus [5], these chemical findings confirm the botanical identification of *M. occidentalis* and further indicate a close relationship with other species of this genus. Additionally, ellagic acid derivatives, which constitute one of the main classes of secondary metabolites isolated from this plant, could then be considered as a chemotaxonomic marker of this species. The anthelmintic potential of schweinfurthins O and B (**9** and **10**) could be investigated as similar compounds (grifolin and geranyl-2-orcinol) showed anthelmintic activity against *C. elegans* and newly transformed schistosomules [42].

## 3. Materials and Methods

### 3.1. General

Electrospray ionization (ESI) mass spectra were recorded on a 1200-series HPLC system or a 1260-series Infinity II HPLC-system (Agilent Technologies, Santa Clara, CA, USA) with a binary pump and integrated diode array detector coupled to an LC/MSDTrap-XTC-mass spectrometer (Agilent Technologies) or an LC/MSD Infinity Lab LC/MSD (G6125B LC/MSD). High-resolution mass spectra were recorded on a Micromass-Q-TOFUltima-3-mass spectrometer (Waters, Milford, MA, USA) with a Lock Spray-interface and a suitable external calibrant. UV—Vis spectra were recorded on an Evolution 201 UV—Visible Spectrophotometer (Thermo Scientific, Waltham, MA, USA), and infrared (IR) spectra were recorded on a Tensor 27 FTIR-spectrometer (Bruker, Billerica, MA, USA) equipped with a diamond ATR. 1D and 2D-NMR spectra were obtained on a Bruker Avance III 500 HD or Avance 600. (Bruker, Bremen, Germany), and TMS was used as an internal standard. Column chromatography was carried out on silica gel 230–400 mesh and silica gel 70–230 mesh (Merck, Darmstadt, Germany). Thin-layer chromatography (TLC) was performed on Merck precoated silica gel 60 F_254_ aluminum foil and revealed using a UV lamp (254–365 nm) and 10% H_2_SO_4_ reagent, followed by heating.

### 3.2. Plant Material

The leaves of *M. occidentalis* were collected at Batoufam village (5°16′42″ north, 10°27′57″ east) near Bandjoun in the Koung-Khi Subdivision, West Region of Cameroon in 2017. Specimens of the collection were deposited at the Cameroon National Herbarium in Yaoundé (Ref no. 50436/NHC).

### 3.3. Extraction and Isolation

The air-dried and powdered leaves of *M. occidentalis* (3.0 kg) were extracted with DCM-MeOH (15 L, 1:1, *v/v*, 3 × 24 h) at room temperature, and the filtrate obtained was evaporated under vacuum at 45 °C to yield the crude DCM-MeOH extract (203.7 g). Part of the extract (200 g) was suspended in distilled water (600 mL) and successively partitioned with EtOAc and n-BuOH (1 L each) to give 76.8 g and 48.5 g of fractions, respectively. Part of the EtOAc fraction (70 g) was subjected to silica gel column chromatography (CC) using gradient elution with the mixture n-hexane/acetone as the mobile phase in a step-gradient from 9:1 to 1:9 (*v/v*) and acetone–MeOH (10:0 to 4:1, *v/v*) to afford four major subfractions (Fr. A−Fr. D). Fr. A (8.5 g) was fractionated by n-hexane–acetone (19:1 to 9:1, *v/v*) on normal-phase CC to give two main subfractions Fr. A.1−Fr. A.2. Lupeol (**4**; 15.2 mg), β-sitosterol, and stigmasterol (**1 + 2**; 25.8 mg) were obtained after the filtration of Fr. A.1 (1.28 g) and Fr. A.2 (3.5 g), respectively. CC of Fr. B (11.6 g) eluted with n-hexane–acetone (17:3, *v/v*) afforded two main subfractions Fr. B.1 (3.2 g) and Fr. B.2 (5.3 g). CC over silica gel of Fr. B.1 eluted with n-hexane–acetone (9:1 to 17:3, *v/v*) led to the isolation of 3′,4′-methylenedioxy-3-O-methylellagic (**12**; 6.9 mg) and 3,3′,4′-tri-O-methyl ellagic acid (**14**; 3.3 mg). Fr. B.2 was subjected to CC over silica gel eluted with n-hexane–acetone (4:1, *v/v*) to afford kaempferol (**7**; 6.2 mg) and quercetin (**8**; 8.1 mg). Fr C (17.8 g) was subjected to CC over Sephadex LH-20 and eluted with DCM–MeOH (1:4, *v/v*) to yield (5R,6R)-4,6-dihydrocarbonyl-5-[2′,3′,4′-trihydroxy-6′-(methoxycarbonyl)phenyl]-5,6-dihydro-2H-pyran-2-one (**15**; 4.5 mg) and isomacarangin (**6**; 57.9 mg). Fr. D (21 g) was separated by CC over silica gel, and eluted with an isocratic system n-hexane–acetone (3:2, *v/v*) to give two main subfractions Fr. D.1 (3.7 g) and Fr. D.2 (4.2 g). Fr. D.1 was repeatedly chromatographed on silica gel with n-hexane–acetone (3:1, *v/v*) to afford schweinfurthin O (**9**; 3.5 mg) and schweinfurthin B (**10**; 25.3 mg). The Sephadex LH-20 CC of Fr D.2 (17.8 g) eluted with MeOH afforded methyl brocchllin carboxylate (**16**; 7.5 mg). The n-BuOH soluble fraction (40 g) was subjected to CC over silica gel eluted with acetone–MeOH (1:0 to 1:1, *v/v*) to give two major fractions Fr. E (12 g) and Fr. F (8.3 g). Fr. E was subjected to CC eluted with the ternary system EtOAc–MeOH–H_2_O (9:1:0.5, *v/v/v*) on normal phase CC to give three subfractions Fr. E. 1−Fr. E.3. Subfraction Fr. E.1 (1.1 g) was subjected to repeated CC eluted with EtOAc–MeOH–H_2_O (95:5:2, *v/v/v*) to afford ishigoside (**17**; 5.2 mg). β-Sitosterol 3-*O*-β-D-glucopyranoside (**3**; 12 mg) precipitated from subfraction Fr. E.2 (2.2 g). Fr. E.3 (1.5 g) was subjected to CC eluted with EtOAc–MeOH–H_2_O (9:1:0.5, *v/v/v*) to afford 3,3′,4-tri-O-methylellagic acid 4′-O-β-D-glucopyranoside (**13**; 8.2 mg). Silica gel CC of Fr. F eluted with the ternary system EtOAc–MeOH–H_2_O (9:1:0.5, *v/v/v*) on normal phase gave two subfractions Fr. F. 1 (2.3 g) and Fr. F.2 (1.7 g). Sephadex LH-20 CC of Fr. F.1 eluted with MeOH afforded ellagic acid (**11**; 9.7 mg). CC of Fr.F.2 eluted with EtOAc–MeOH–H_2_O (9:0.7:0.3, *v/v/v*) afforded apigenin-7-O-β-D-glycopyranoside (**5**; 4.6 mg). A simplified scheme describing the extraction, fractionation, and isolation of compounds 1–17 is presented in the Appendix A.

### 3.4. Spectroscopic Data of the Isolated Compounds

*β*-Sitosterol (**1**): white powder; ^1^H-NMR (500 MHz, CDCl_3_): δ_H_ (mult, *J* in Hertz): 5.34 (m, H-6), 3.52 (m, H-3), 1.01 (s, H-27), 0.92 (d, *J* = 6.5 Hz, H-19, H-21), 0.84 (d, *J* = 2.9 Hz, H-29), 0.83 (d, *J* = 2.2 Hz, H-26), 0.68 (s, H-18); ^13^C NMR (125 MHz, CDCl_3_): δ_C_ 140.8 (C-5), 121.8 (C-6), 71.9 (C-3), 56.9 (C-14), 56.2 (C-17), 50.2 (C-9), 45.9 (C-24), 42.7 (C-13), 42.4 (C-4), 39.9 (C-12), 37.4 (C-1), 36.6 (C-10), 36.2 (C-20), 34.0 (C-22), 32.1 (C-7), 32.0 (C-8), 31.8 (C-2), 29.2 (C-25), 28.4 (C-16), 26.2 (C-23), 24.4 (C-15), 23.2 (C-28), 21.2 (C-11), 19.9 (C-26), 19.5 (C-27), 19.1 (C-19), 18.9 (C-21), 12.1 (C-18), 12.0 (C-29).

Stigmasterol (**2**): white powder; ^1^H-NMR (500 MHz, CDCl_3_): δ_H_ (mult, *J* in Hertz): 5.34 (m, H-6), 5.15 (dd, *J* = 15.1, 8.7 Hz, H-22), 5.01 (dd, *J* = 15.1, 8.7 Hz, H-23), 3.52 (m, H-3), 1.25 (brs, H-19), 1.01 (s, H-27), 0.94 (brs, H-21), 0.84 (d, *J* = 2.9 Hz, H-29), 0.81 (s, H-24), 0.70 (s, H-18; ^13^C NMR (125 MHz, CDCl_3_): δ_C_ 140.8 (C-5), 138.4 (C-22), 129.1 (C-23), 121.8 (C-6), 71.5 (C-3), 57.0 (C-14), 56.1 (C-17), 51.3 (C-9), 45.3 (C-24), 42.7 (C-13), 42.3 (C-4), 40.6 (C-20), 39.8 (C-12), 37.1 (C-1), 36.0 (C-10), 32.7 (C-7), 32.2 (C-8), 31.6 (C-2), 29.0 (C-25), 28.8 (C-16), 25.5 (C-28), 24.5 (C-15), 21.3 (C-11), 20.8 (C-21), 18.8 (C-26), 18.4 (C-27), 17.7 (C-19), 12.4 (C-18), 12.2 (C-29).

*β*-Sitosterol-3-*O*-*β*-D-glucopyranoside (**3**): white powder; ^1^H NMR (500 MHz, C_5_D_5_N, 25 °C, TMS): 5.34 (brs, 1H, H-6), 5.03 (d, *J* = 7.5 Hz, 1H, H-1′), 4.56 (dd, *J* = 11.9, 2.6 Hz, 2H, H-6′), 4.28 (m, H-4′, H-5′), 4.05 (t, 8.1 Hz, H-2′), 3.96 (m, H-3, H-3′), 1.42 (m, H-27), 1.28 (br s, H-28), 0.97 (d, *J* = 6.3 Hz, H-20, H-24, H-26), 0.92 (s, H-19), 0.87 (br s, H-29), 0.84 (br s, H-21); ^13^C NMR (125 MHz, C_5_H_5_N): δ_C_ 37.6 (C-1), 30.4 (C-2), 78.8 (C-3), 40.1 (C-4), 141.1 (C-5), 122.1 (C-6), 32.2 (C-7), 32.3 (C-8), 50.5 (C-9), 37.1 (C-10), 21.4 (C-11), 39.5 (C-12), 42.6 (C-13), 56.4 (C-14), 24.7 (C-15), 28.6 (C-16), 57.0 (C-17), 12.3 (C-18), 19.6 (C-19), 36.5 (C-20), 19.4 (C-21), 34.4 (C-22), 26.5 (C-23), 46.2 (C-24), 29.6 (C-25), 19.2 (C-26), 20.1 (C-27), 23.5 (C-28), 12.1 (C-29), 102.7 (C-1′), 75.5 (C-2′), 78.6 (C-3′), 71.8 (C-4′), 78.3 (C-5′), 63.0 (C-6′).

Lupeol (**4**): ^1^H NMR (500 MHz, CDCl_3_, 25 °C, TMS): δ = 4.69 (d, *J* = 2.4 Hz, 1H, H-29b), 4.56 (d, *J* = 2.7 Hz, 1H, H-29a), 3.18 (dd, *J* = 11.4, 4.9 Hz, 1H, H-3), 1.68 (s, 3H, H-30), 1.03 (s, 3H, H-26), 0.97 (s, 3H, H-23), 0.94 (s, 3H, H-27), 0.83 (s, 3H, H-25), 0.79 (s, 3H, H-28). ^13^C NMR (125 MHz, CDCl_3_): 38.8 (C-1), 27.3 (C-2), 79.1 (C-3), 39.0 (C-4), 55.4 (C-5), 18.4 (C-6), 34.3 (C-7), 40.9 (C-8), 50.6 (C-9), 37.3 (C-10), 21.0 (C-11), 25.2 (C-12), 38.2 (C-13), 42.9 (C-14), 27.5 (C-15), 35.7 (C-16), 43.1 (C-17), 48.4 (C-18), 48.1 (C-19), 151.0 (C-20), 29.9 (C-21), 40.1 (C-22), 28.0 (C-23), 14.7 (C-24), 16.2 (C-25), 16.1 (C-26), 15.5 (C-27), 18.1 (C-28), δ_C_ 109.4 (C-29), 19.4 (C-30).

Apigenin-7-*O*-*β*-D-glycoside (**5**): ^1^H NMR (500 MHz, C_5_D_5_N, 25 °C, TMS): δ = 13.60 (d, *J* = 4.2 Hz, 1H), 6.90 (brs,1H, H-3), 6.84 (brs, 1H, H-6), 7.09 (brs, 1H, H-8), 7.90 (d, *J* = 8.7 Hz, 2H, H-2′/H-6′), 7.17 (d, *J* = 8.7 Hz, 2H, H-3′/H-5′), 5.84 (d, *J* = 5.6 Hz, 1H, H-1″), 4.35 (dd, *J* = 5.6, 9.5 Hz 1H, H-2″), 4.22 (m, 1H, H-3″), 4.35 (m, 1H, H-4″), 4.56/4.37(d, *J* = 11.9 Hz, 2H, H-6_a,b_″), 4.40 (m, 1H, H-5″).—^13^C NMR (125 MHz, C_5_D_5_N): δ = 165.4 (C-2), 104.4 (C-3), 183.4 (C-4), 163.0 (C-5), 101.2 (C-6), 164.5 (C-7), 95.7(C-8), 158.2 (C-8a), 107.0 (C-4a), 122.4 (C-1′), 129.5(C-2′/C-6′), 117.4 (C-3′/C-5′), 163.3 (C-4′), 102.1 (C-1″), 75.2 (C-2″), 78.8 (C-3″), 71.5 (C-4″), 79.7(C-5″), 62.7(C-6″).

Isomacarangin (**6**): ^1^H NMR (500 MHz, CD_3_OD, 25 °C, TMS): δ = 6.21 (s, 1H, H-6), 8.08 (d, *J* = 8.9 Hz, 2H, H-2′/H-6′), 6.86 (d, *J* = 8.9 Hz, 2H, H-3′/H-5′), 3.48 (d, *J* = 6.8 Hz, 2H, H-1″), 5.19 (m, 1H, H-2″), 1.94 (t, *J* = 7.3 Hz, 2H, H-4″), 2.00 (t, *J* = 7.3 Hz, 2H, H-5″), 4.96 (m, 1H, H-6″), 1.77 (s, 3H, H-8″), 1.49 (d, *J* = 1.5 Hz, 3H, H-9″), 1.44 (s, 3H, H-10″)—^13^C NMR (125 MHz, CD_3_OD) δ = 147.8 (C-2), 136.8 (C-3), 177.4 (C-4), 155.4 (C-5), 98.5 (C-6), 162.5 (C-7), 107.5 (C-8), 159.9 (C-8a), 104.3 (C-4a), 123.9 (C-1′), 130.6 (C-2′), 116.1 (C-3′), 160.4 (C-4′), 116.1 (C-5′), 130.6 (C-6′), 22.2 (C-1″), 123.9 (C-2″), 135.9 (C-3″), 40.5 (C-4″), 27.4 (C-5″), 125.1 (C-6″), 132.0 (C-7″), 16.4 (C-8″), 25.6 (C-9″), 17.3 (C-10″).

Kaempferol (**7**): ^1^H NMR (500 MHz, acetone-*d_6_*, 25 °C, TMS): δ = 6.55 (d, *J* = 1.8 Hz, 1H, H-6), 6.29 (d, *J* = 1.8 Hz, 1H, H-8), 7.03 (d, *J* = 8.8 Hz, 2H, H-2′/H-6′), 8.17 (d, *J* = 8.8 Hz, 2H, H-3′/H-5′). ^13^C NMR (125 MHz, acetone- *d_6_*) δ = 146.1 (C-2), 137.0 (C-3), 156.9 (C-5), 93.6 (C-6), 164.2 (C-7), 98.3 (C-8), 161.6 (C-8a), 103.5 (C-4a), 122.9 (C-1′), 129.5 (C-2′/C-6′), 115.4 (C-3′/C-5′), 159.2 (C-4′).

Quercetin (**8**): ^1^H NMR (500 MHz, DMSO–*d_6_*, 25 °C, TMS): δ = 6.52 (d, *J* = 1.9 Hz, 1H, H-6), 6.26 (d, *J* = 1.9 Hz, 1H, H-8), 6.99 (d, *J* = 8.5 Hz, 1H, H-2′), 7.82 (d, *J* = 2.1 Hz, 1H, H-5′), 7.69 (dd, *J* = 8.5, 2.1 Hz, 1H, H-6′).—^13^C NMR (125 MHz, DMSO–*d_6_*) δ = 146.8 (C-2), 136.6 (C-3), 176.4 (C-4), 157.7 (C-5), 94.4 (C-6), 162.3 (C-7), 99.0 (C-8), 162.0 (C-8a), 104.1 (C-4a), 123.7 (C-1′), 116.1 (C-2′), 145.7 (C-3′), 148.2 (C-4′), 115.6 (C-5′), 121.4 (C-6′).

Schweinfurthin O (**9**): ^1^H NMR (500 MHz, acetone- *d_6_*, 25 °C, TMS): δ = 7.03 (d, *J* = 2.0 Hz, 1H, H-3), 6.87 (d, *J* = 2.0 Hz, 1H, H-5), 6.80 (s, 1H, H-6), 6.81 (d, *J* = 16.2 Hz, 1H, H-1′), 6.76 (d, *J* = 16.2 Hz, 1H, H-2′), 6.57 (s, 1H, H-4′), 6.57 (s, 1H, H-8′), 3.37 (d, *J* = 7.1 Hz, 2H, H-1″), 5.33 (tq, *J* = 7.2, 1.4 Hz, 1H, H-2″), 1.79 (d, *J* = 1.3 Hz, 3H, H-4″), 1.98 (s, 1H, H-5″), 2.06 (d, *J* = 1.3 Hz, 2H, H-6″), 5.12 (ddt, *J* = 7.1, 4.0, 1.4 Hz, 1H, H-7″), 1.57 (s, 3H, H-9″), 1.95 (s, 1H, H-10″), 2.02 (d, *J* = 7.1 Hz, 2H, H-11″), 5.08 (m, 1H, H-12″), 1.57 (s, 3H, H-14″), 1.63 (dd, *J* = 5.2, 1.4 Hz, 3H, H-15″)—^13^C NMR (125 MHz, acetone- *d_6_*) δ = 146.2 (C-1), 145.0 (C-2), 113.7 (C-3), 136.3 (C-4), 119.9 (C-5), 116.3 (C-6), 128.4 (C-1′), 127.0 (C-2′), 130.9 (C-3′), 105.7 (C-4′), 156.1 (C-5′), 115.3 (C-6′), 156.9 (C-7′), 105.7 (C-8′), 23.2 (C-1″), 124.3 (C-2″), 134.6 (C-3″), 16.3 (C-4″), 40.6 (C-5″), 27.4 (C-6″), 125.1 (C-7″), 134.3 (C-8″), 16.1 (C-9″), 40.5 (C-10″), 27.5 (C-11″), 125.3 (C-12″), 131.6 (C-13″), 17.7 (C-14″), 25.9 (C-15″).

Schweinfurthin B (**10**): [α]_D_ = +44.5° (c 1.0, EtOH); ^1^H NMR (500 MHz, acetone- *d_6_*, 25 °C, TMS): δ = 3.30 (d, *J* = 3.5 Hz, 1H, H-2), 4.13(m, 1H, H-3), 2.29/1.91 (dd, J = 13.9, 3.1 Hz, 2H, H-4_a,b_), 6.92 (d, J = 1.9 Hz, 1H, H-6), 6.81 (d, *J* = 1.9 Hz, 1H, H-8), 2.75 (m, 2H, H-9), 1.68 (dd, *J* = 12.7, 5.3 Hz, 1H, H-9a), 1.05 (s, 6H, H-11/H-12), 1.36 (s, 3H, H-13), 6.88 (m, 2H, H-1′/H-2′), 6.57 (s, 2H, H-4′/H-8′), 3.35 (d, *J* = 7.1 Hz, 2H, H-1″), 5.30 (m, 1H, H-2″), 1.75 (d, *J* = 1.4 Hz, 3H, H-4″), 1.91 (d, *J* = 7.2 Hz, 2H, H-5″), 2.05 (s, 1H, H-6″), 5.05 (m, 1H, H-7″), 1.52 (d, *J* = 1.3 Hz, 3H, H-9″), 1.58 (d, *J* = 1.5 Hz, 3H, H-10″), 3.75 (s, 3H, 5-OCH_3_).—^13^C NMR (125 MHz, acetone- *d_6_*) δ = 38.3 (C-1), 77.6 (C-2), 71.3 (C-3), 44.0 (C-4), 76.8 (C-4a), 149.7 (C-5), 107.9 (C-6), 129.8 (C-7), 121.0 (C-8), 123.4 (C-8a), 22.7 (C-9), 47.5 (C-9a), 143.1 (C-10a), 16.1 (C-11), 28.9 (C-12), 21.7 (C-13), 128.1 (C-1′), 126.7 (C-2′), 136.9 (C-3′), 105.4 (C-4′/C-8′), 156.6 (C-5′), 114.8 (C-6′), 156.6 (C-7′), 23.3 (C-1″), 123.7 (C-2″), 134.1 (C-3″), 15.9 (C-4″), 40.2 (C-5″), 27.1 (C-6″), 124.8 (C-7″), 136.1 (C-8″), 17.3 (C-9″), 25.5 (C-10″), 55.7 (5-OCH_3_).

Ellagic acid (**11**): ^1^H NMR (500 MHz, DMSO-*d_6_*, 25 °C, TMS): δ = 7.81 (s, 2H, H-2/H-2′)—^13^C NMR (125 MHz, DMSO-*d_6_*) δ = 108.6 (C-1/C-1′), 111.5 (C-2/C-2′), 149.6 (C-3/C-3′), 145.1 (C-4/C-4′), 137.7 (C-5/C-5′), 113.6 (C-6/C-6′), 160.6 (C-7/C-7′).

3′,4′-Methylenedioxy-3-*O*-methylellagic (**12**): ^1^H NMR (500 MHz, DMSO-*d_6_*, 25 °C, TMS): δ = 7.56 (s, 1H, H-5), 7.55 (s, 1H, H-5′), 4.06 (s, 3H, 3-OCH_3_), 6.40 (s, 2H, -OCH_2_-).—^13^C NMR (125 MHz, DMSO*-d_6_*) δ = 112.7 (C-1), 131.1 (C-2), 140.2 (C-3), 152.7 (C-4), 112.0 (C-5), 116.1 (C-6), 157.7 (C-7), 111.2 (C-1′), 141.6 (C-2′), 138.3 (C-3′), 150.0 (C-4′), 103.9 (C-5′), 111.0 (C-6′), 158.3 (C-7′), 61.0 (3-OMe), 104.1 (-OCH_2_O-).

3,3′,4-Tri-*O*-methylellagic acid 4′-*O*-*β*-*D*-glucopyranoside (**13**): ^1^H NMR (500 MHz, C_5_D_5_N, 25 °C, TMS): δ = 7.85 (d, *J* = 1.7 Hz, 1H, H-5) 8.49 (d, *J* = 1.7 Hz, 1H, H-5′), 5.94 (d, *J* = 5.6 Hz, 1H, H-1′), 4.45 (s, 1H, H-2′), 4.44 (s, 1H, H-3′), 4.42 (d, *J* = 1.7 Hz, 1H, H-4′), 4.21 (d, *J* = 1.9 Hz, 1H, H-5′), 4.62 (d, *J* = 12.1 Hz, 1H, H-6′), 4.17 (d, *J* = 1.7 Hz, 3H, 3-OCH_3_), 3.89 (d, *J* = 1.7 Hz, 3H, 4-OCH_3_), 4.30 (d, *J* = 1.7 Hz, 3H, 3′-OCH_3_).—^13^C NMR (125 MHz, C_5_D_5_N): δ = 113.7 (C-1), 142.5(C-2), 142.4 (C-3), 155.5 (C-4), 108.5 (C-5), 114.6 (C-6), 159.3 (C-7), 114.0 (C-1′), 143.3 (C-2′), 142.5 (C-3′), 153.5 (C-4′), 113.8 (C-5′), 113.9 (C-6′), 159.5 (C-7′), 103.4 (C-1″), 75.3 (C-2″), 79.0 (C-3″), 71.5 (C-4″), 79.6 (C-5″), 62.8 (C-6″), 62.0 (3-OCH_3_), 57.1(4-O CH_3_), 62.4 (3′-OCH_3_).

3,3′,4′-Tri-*O*-methyl ellagic acid (**14**): ^1^H NMR (500 MHz, C_5_D_5_N, 25 °C, TMS): δ = 8.08 (s, 1H, H-5), 7.86 (s, 1H, H-5′), 4.17 (s, 3H, 3-OCH_3_), 4.23 (s, 3H, 3′-OCH_3_), 3.88 (s, 3H, 4′-OCH_3_).—^13^C NMR (125 MHz, C_5_D_5_N) δ = 114.1 (C-1), 142.1 (C-2), 141.9 (C-3), 154.7 (C-4), 108.2 (C-5), 113.2 (C-6), 159.5 (C-7), 113.5 (C-1′), 142.5 (C-2′), 141.5 (C-3′), 154.5 (C-4′), 113.1 (C-5′), 112.0 (C-6′), 159.5 (C-7′), 61.7 (3-OCH_3_), 61.5 (3′-OCH_3_), 56.8 (4′-OCH_3_).

(5*R*,6*R*)-4,6-Dihydrocarbonyl-5-[2′,3′,4′-trihydroxy-6′(methoxycarbonyl)phenyl]-5,6-dihydro-2H-pyran-2-one (**15**): [α]_D_ = +171.2° (c 0.5, acetone); ^1^H NMR (500 MHz, CD_3_OD, 25 °C, TMS): δ = 6.85 (s, 1H, H-3), 5.38 (d, *J* = 1.9 Hz, 1H, H-5), 5.32 (d, *J* = 1.9 Hz, 1H, H-6), 7.06 (br s, 1H, H-5′), 3.66 (s, 3H, 4-CH_3_), 3.71 (s, 3H, 6-CH_3_), 3.66 (s, 3H, 6′-CH_3_).—^13^C NMR (125 MHz, CD_3_OD): δ = 166.9 (C-2), 130.3 (C-3), 143.1 (C-4), 35.7 (C-5), 80.3 (C-6), 117.1 (C-1′), 144.3 (C-2′), 140.5 (C-3′), 147.0 (C-4′), 108.8 (C-5′), 116.3 (C-6′), 167.8 (1″-C=O), 53.5 (1″-OCH_3_), 171.1 (2″-C=O), 52.7 (2″-OCH_3_), 165.4 (3″-C=O), 53.1 (3″-OCH_3_).

Methylbrocchllin carboxylate (**16**): ^1^H NMR (500 MHz, CD_3_OD, 25 °C, TMS): δ = 2.89/2.41 (dd, *J* = 18.7, 7.7 Hz, 1H, H-2_a.b_), 4.35 (dd, *J* = 7.7, 2.1 Hz, 1H, H-3), 7.22 (s, 1H, H-6), 3.54 (s, 3H, OCH_3_).—^13^C NMR (125 MHz, CD_3_OD): δ = 202.7 (C-1), 46.6 (C-2), 50.2 (C-3), 155.5 (C-3a), 169.5 (C-5), 122.7 (C-5a), 117.8 (C-6), 159.3 (C-7), 150.0 (C-8), 153.2 (C-9), 124.7 (C-9a), 148.1 (C-3b), 182.2 (3-C=O), 61.7 (OCH_3_).

Ishigoside (**17**): ^1^H NMR (500 MHz, CDCl_3_/CD_3_OD, 25 °C, TMS): δ = 4.20/3.70 (d, *J* = 5.3 Hz, 2H, H-1_a,b_), 5.44 (m, 1H, H-2), 4.61/4.30 (dd, *J* = 12.1, 3.0 Hz, 2H, H-3_a,b_), 4.90 (d, *J* = 3.8 Hz, 1H, H-1′), 3.55 (dd, *J* = 9.7, 3.8 Hz, 1H, H-2′), 3.76 (t, *J* = 9.3 Hz, 1H, H-3′), 3.28 (t, *J* = 9.4 Hz, 1H, H-4′), 4.17 (d, *J* = 2.0 Hz, 1H, H-5′), 3.46/3.10 (dd, *J* = 14.4, 8.6 Hz 1H, H-6′_a,b_), 2.45 (m, 4H, H-2″/H-2‴), 1.38 (s, 24H, H-3″-H-14″, H-3‴-H-14‴), 1.71 (m, 4H, H-15″/H-15‴), 0.99 (t, *J* = 7.0 Hz, 6H, H-16″/H-16‴).—^13^C NMR (125 MHz, CDCl_3_/CD_3_OD) δ = 66.7 (C-1), 71.2 (C-2), 63.9 (C-3), 99.5 (C-1′), 72.8 (C-2′), 74.4 (C-3′), 74.3 (C-4′), 69.3 (C-5′), 53.8 (C-6′), 175.0 (C-1″), 34.9 (C-2″), 32.6–29.8 (C-3″-C-14″), 25.6 (C-15″), 174.8 (C-1‴), 34.8 (C-2‴), 32.6–29.8 (C-3‴-C-14‴), 23.6 (C-15‴), 14.7 (C-16″/C-16‴).

### 3.5. Preparation of the Semisynthetic Derivatives

#### 3.5.1. Allylation of Isomacarangin (**6**)

Isomacangin (**6**) (25.1 mg, 0.059 mmol) was dissolved in 3 mL of acetone; allyl bromide (3 mL) and K_2_CO_3_ (5 mg, 0.036 mmol) were added successively. The mixture was magnetically stirred at 25 °C and monitored by TLC until the disappearance of the starting material [43]. After 24 h of reaction, it was poured into ice (100 g) and extracted with EtOAc (3 × 10 mL). The organic layer was washed with water (3 × 20 mL), dried over Na_2_SO_4_, and evaporated. The crude product was purified by adsorptive filtration on silica gel (short column, *n*-hexane-acetone 19:1, *v/v*) to afford 3,7,4′-triallylisomacarangin (**6a**) (12.6 mg, 37.4%) and 3,6,7,4′-tetraallylisomacarangin (**6b**) (6.5 mg, 17.4%) as a yellowish powder.

3,7,4′-Triallylisomacarangin (**6a**): ^1^H NMR (500 MHz, acetone- *d_6_*, 25 °C, TMS): δ = 6.74 (s, 1H, H-6), 8.13 (d, *J* = 9.0 Hz, 2H, H-2′/H-6′), 7.12 (d, *J* = 9.0 Hz, 2H, H-3′/H-5′), 3.38 (d, *J* = 7.2 Hz, 2H, H-1″), 5.26 (d, *J* = 1.5 Hz, 1H, H-2″), 1.96 (dd, *J* = 9.1, 6.3 Hz, 2H, H-4″), 2.05 (s, 2H, H-5″), 5.07 (m, 1H, H-6″), 1.55 (s, 3H, H-8″), 1.60 (d, *J* = 1.7 Hz, 3H, H-9″), 1.79 (d, *J* = 1.3 Hz, 3H, H-10″), 3-O-allyl [4.55 (d, *J* = 6.0 Hz, 2H), 6.00 (m, 1H), 5.30 (m, 2H) ], 7-O-allyl [4.75 (d, *J* = 6.0, Hz, 2H), 5.16 (m, 1H), 5.50 (m, 2H) ], 4′-O-allyl [4.69 (d, *J* = 6.0, Hz, 2H), 6.13 (m, 1H), 5.50 (m, 2H)],—^13^C NMR (125 MHz, acetone- *d_6_*) δ = 156.7 (C-2), 138.2 (C-3), 179.3 (C-4), 156.0 (C-5), 91.7 (C-6), 163.0 (C-7), 113.0 (C-8), 158.6 (C-8a), 106.4 (C-4a), 123.9 (C-1′), 131.1 (C-2′/C-6′), 115.4 (C-3′/C-5′), 161.6 (C-4′), 22.0 (C-1″), 122.9 (C-2″), 135.5 (C-3″), 40.2 (C-4″), 27.3 (C-5″), 125.1 (C-6″), 131.6 (C-7″), 17.8 (C-8″), 25.7 (C-9″), 16.3 (C-10″), 3-O-allyl [73.7 (CH_2_), 134.8 (CH), 118.3 (CH_2_)], 7-O-allyl [70.0 (CH_2_), 133.8 (CH), 117.8 (CH_2_)], 4′-O-allyl [69.5 (CH_2_), 134.2 (CH), 117.9 (CH_2_)]; HRESIMS [M + Na]^+^ at *m/z* 565.2560 (calcd. *m/z* 565.2560 for C_34_H_38_O_6_Na^+^).

3,6,7,4′-Tetraallylisomacarangin (**6b**): ^1^H NMR (500 MHz, acetone- *d_6_*, 25 °C, TMS): δ = 8.17 (d, *J* = 9.0 Hz, 2H, H-2′/H-6′), 7.14 (d, *J* = 9.0 Hz, 2H, H-3′/H-5′), 3.41 (d, *J* = 6.8 Hz, 2H, H-1″), 5.28 (m, 1H), 1.93 (t, *J* = 7.3 Hz, 2H, H-4″), 2.00 (t, *J* = 7.3 Hz, 2H, H-5″), 5.06 (s, 1H, H-6″), 1.54 (s, 3H, H-8″), 1.59 (d, *J* = 1.5 Hz, 3H, H-9″), 1.77 (s, 3H, H-10″), 3-O-allyl [4.75 (d, *J* = 6.0 Hz, 2H), 5.16 (m, 1H), 5.50 (m, 2H) ], 6-C-allyl [3.64 (m, 2H), 6.05 (m, 1H), 7.14 (m, 2H)], 7-O-allyl [4.45 (d, *J* = 6.0, Hz, 2H), 5.90 (m, 1H), 5.26 (m, 2H) ], 4′-O-allyl [4.70 (d, *J* = 6.0, Hz, 2H), 6.12 (m, 1H), 5.17 (m, 2H)]—^13^C NMR (125 MHz, acetone- *d_6_*) δ = 158.3 (C-2), 138.2 (C-3), 178.7 (C-4), 157.3 (C-5), 112.3 (C-6), 162.1 (C-7), 112.3 (C-8), 153.2 (C-8a), 108.7 (C-4a), 124.0 (C-1′), 131.3 (C-2′/C-6′), 115.6 (C-3′/C-5′), 161.8 (C-4′), 23.1 (C-1″), 123.5 (C-2″), 135.8 (C-3″), 39.6 (C-4″), 27.2 (C-5″), 125.0 (C-6″), 131.7 (C-7″), 17.4 (C-8″), 25.8 (C-9″), 16.4 (C-10″), 3-O-allyl [73.7 (CH_2_), 134.8 (CH), 118.4 (CH_2_)], 6-C-allyl [28.5 (CH_2_), 137.3 (CH), 115.7 (CH_2_)], 7-O-allyl [76.1 (CH_2_), 134.7 (CH), 117.4 (CH_2_)], 4′-O-allyl [69.4 (CH_2_), 134.2 (CH), 117.9 (CH_2_)]; HRESIMS [M + H]^+^ at *m/z* 583.3052 (calcd. *m/z* 583.3054 for C_37_H_43_O_6_^+^).

#### 3.5.2. Acetylation of Isomacarangin (**6**)

Pyridine (5 mL) was added to a powder of isomacarangin (**6**) (25.1 mg, 0.059 mmol) and then acetic anhydride (3 mL) was added. The mixture was magnetically stirred at 25 °C and monitored by TLC until the disappearance of the starting material. After 24 h of reaction, it was poured into ice (100 g) and extracted with EtOAc (3 × 10 mL). The organic layer was washed with water (3 × 20 mL), dried over Na_2_SO_4_, and evaporated. The crude product was purified by adsorptive filtration on silica gel (short column, *n*-hexane–acetone 17:3, *v/v*) to yield 3,5,7,4′-tetraacetylisomacarangin (**6c**) (34.9 mg, 100%) as a yellowish powder.

3,5,7,4′-Tetraacetylisomacarangin (**6c**): ^1^H-NMR (500 MHz, acetone- *d_6_*, 25 °C, TMS): δ = 8.31 (s, 1H, H-6), 8.81 (d, *J* = 8.8 Hz, 2H, H-2′/H-6′), 8.16 (d, *J* = 8.8 Hz, 2H, H-3′/H-5′), 4.13 (br s, 2H, H-1″), 5.83 (m, 1H, H-2″), 2.76/2.84 (m, 2H, H-4″), 2.83 (m, 2H, H-5″), 5.86 (m, 1H, H-6″), 2.58 (s, 3H, H-8″), 2.42 (s, 3H, H-9″), 2.36 (s, 3H, H-10″), 2.31 (3-CH_3_C=O), 2.32 (5-CH_3_C=O), 2.39 (7-CH_3_C=O), 2.40 (4′-CH_3_C=O).—^13^C NMR (125 MHz, acetone- *d_6_*) δ = 154.3 (C-2), 134.6 (C-3), 170.5 (C-4), 149.0 (C-5), 111.7 (C-6), 155.2 (C-7), 126.4 (C-8), 155.6 (C-8a), 115.4 (C-4a), 127.9 (C-1′), 130.6 (C-2′/C-6′), 123.2 (C-3′/C-5′), 154.2 (C-4′), 23.8 (C-1″), 121.6 (C-2″), 136.9 (C-3″), 40.3 (C-4″), 27.2 (C-5″), 124.9 (C-6″), 131.8 (C-7″), 17.7 (C-8″), 25.7 (C-9″), 16.4 (C-10″), 20.4/169.4 (3-CH_3_C=O), 21.1/168.3 (5-CH_3_C=O), 20.4/169.2 (7-CH_3_C=O), 21.0/168.4 (4′-CH_3_C=O); HRESIMS [M + Na]^+^ at *m/z* 613.2043 (calcd. *m/z* 613.2044 for C_33_H_34_O_10_Na^+^).

### 3.6. Antimicrobial Assays

#### 3.6.1. Antibacterial Activity

The screenings were performed in duplicate three times in sterile 96 well microplates. Indeed, 98 μL and 95 μL of MHB culture medium were introduced into the first wells corresponding to the extracts and compounds, respectively, and 50 μL was introduced into the rest of the wells. Subsequently, 2 μL of a sterile solution of extracts concentrated at 100 mg/mL and 5 μL of a solution of compounds concentrated at 20 mg/mL were taken and introduced into the corresponding wells followed by a serial of four dilutions of geometric order 2. Finally, 50 μL of a bacterial suspension at a load of 10^6^ cells/mL was distributed in the test wells and those of the negative control. The concentrations of extract, fractions, and compounds in wells ranged from 1000 μg/mL to 62.5 μg/mL, 500 µg/mL to 31.25 μg/mL, respectively, and from 0.25 μg/mL to 0.0153 μg/mL, for the ciprofloxacin used as positive controls. The final charge of the inoculum in each well was 5 × 10^5^ cells/mL with 200 μL as the final volume. The sterility control was constituted only of the culture medium. The positive control consisted of the culture medium, inoculum, and ciprofloxacin. The microplates were covered and then incubated at 37 °C for 24 h. At the end of the incubation period, 10 μL of a freshly prepared resazurin solution (0.15 mg/mL) was added to all wells, and the plates were once again incubated under the same conditions for 30 min. The smallest concentration at which there was no change in coloration from blue to pink corresponding to a lack of visible bacterial growth was considered as the MIC.

#### 3.6.2. Antifungal Activity

The screenings were performed in duplicate in sterile 96 well microplates. Indeed, 96 μL and 95 μL of SDB culture medium were introduced into the first wells corresponding to the compounds, fractions, and extracts, respectively, and 50 μL was introduced into the rest of the wells. Subsequently, 5 μL of a sterile solution of compounds concentrated at 20 mg/mL and 4 μL of a sterile solution of extracts concentrated at 100 mg/mL were taken and introduced into the corresponding wells, followed by a serial of five geometric dilutions of order 2. Finally, 50 μL of a fungal suspension at a load of 2 × 10^4^ cells/mL was distributed in the test wells and those of the negative control. Concentrations of extracts, compounds, and fluconazole in the wells ranged from 2000 μg/mL to 62.5 μg/mL, 500 μg/mL to 3.890 μg/mL, and from 1.25 μg/mL to 0.0383 μg/mL, respectively, and the final charge of the inoculum in each well was 10^4^ cells/mL. The sterility control was constituted only of the culture medium. The positive control consisted of the culture medium, inoculum, and fluconazole. The microplates were covered and then incubated at 37 °C for 48 h. At the end of the incubation period, 10 μL of a freshly prepared resazurin solution (0.15 mg/mL) was added to all wells and the plates were once again incubated under the same conditions for 30 min. The smallest concentration at which there was no change in coloration from blue to pink corresponding to a lack of visible fungal growth was considered as the MIC.

## Data Availability

Not applicable.

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
