# Peer review of "Chemical Constituents of Macaranga occidentalis, Antimicrobial and Chemophenetic Studies"

_molecules, 2022, doi:10.3390/molecules27248820_

Round 1

Reviewer 1 Report

The work is interesting. I think minor revision is required to improve the manuscript. I attached the reviewed PDF file. My comments are given below:

1. Line 35-36: MIC, n-BuOH-- use full forms.

2. Line 36, 37, ..: symbol of microgram --- will be in non-italicize from throughout the manuscript.

3. Line 63-67: add more references.

4. Line 72-73: EtOAc, n-BuOH--- use full form in first time.

5. Line 82: Cite Figure 1 here.

6. Line 120: magadula (2014) --- check journal's style

7. Line 126: Salah et al. (2003)--- Use number system

8. Line 148: [31,32,33]--- check journal's style

9. Scheme 2: It will be better if mention the name of the compounds. In ligand, use the word 'red'

10. Mgoumfo et al. 2008 ---- check journal's style

11. Batoafam village--- use latitude and longitude

12. A lot of errors in reference list (see the reviewed PDF), e.g., reference 1, 3, 6, 13, 22, 24, 30, 32, 34, 37, 39 & 40. 

Reviewer 2 Report

The submitted manuscript describes the chemical composition of Macaranga occidentalis leaves (Euphorpbiaceae) and the antimicrobial activity of the extract, fractions, isolated compounds, and some semi-synthetic derivatives. In general terms, the manuscript is well-conducted. I would like the authors to review some points and correct minor text editing.

Line 42: in vitro (in italics)

L. 49: viruses, fungi

L. 61 and throughout the manuscript: do not include the term family next to Euphorbiaceae, as the ending -aceae already informs that the taxonomic level is family.

L.64: . Previous

In acknowledgment: missing to complete the sentence

L. 119-124: The results of antimicrobial activity of methanolic extracts of several Macaranga species were not carried out by Magadula (2014), as the authors mention. To review. The correct authors (cited by Magadula, 2014) are Chan et al., 2007 and Lim et al., 2009.

- Methods:

In: 3.3. Extraction and isolation: I suggest that the authors replace much of the description of compound isolation with a schematic of that isolation. This makes it easy to see all the steps.

In: 3.4. Spectroscopic data of the isolated compounds: review the acronyms of C5D5N. Several missing N.

In: β-Sitosterol-3-O-β-D-glucopyranoside, 13C NMR (125 MHz, CDCl3). Swap CDCl3 for the correct solvent

In: 3.6.1. Antibacterial activity: it was not clear what is the final concentration range of ciprofloxacin in the wells.

- List the species of bacteria and yeasts (with their ATCC) used in antimicrobial activity

- Results and discussion:

Review table numbering

In: 2.1. Isolation of specialized metabolites from M. occidentalis: the authors present the isolated compounds and then repeat all of them in item 2.4, classifying them. I suggest that the authors already classify them in item 2.1, avoiding repetition.

In 2.4. Chemical significance of the isolated compounds:

- The description of chemical significance is exhaustive and in my view is confusing. The authors suggest three groups of markers for: family (glyceride), genus (flavonoids and stilbenes – these are already known) and species (ellagic derivatives).

I agree that flavonoids are very useful as markers, but the question of the pattern/type of substitution has yet to be explored. Magadula, 2014 states that Macaranga species have mainly flavonoids with terpene substituents. Can you show in which other Euphorbiaceae genus there is a predominance of this type of flavonoid?

I think it's dangerous for you to suggest glycerides as a marker for Euphorbiaceae if this type of compound has not been isolated from other genus.

The part of the discussion where you suggest the relationship between the genera Macaranga, Phyllanthus and Chrozophora is very confusing. Phyllanthus no longer belongs to Euphorbiaceae. Furthermore, the authors base this relationship on a geraniinic acid derivative and a coumarin with the third ring coming from the malonate pathway. They do not report whether structurally analogous compounds were isolated from Phyllanthus and Chrozophora. How to make this suggestion if most of the coumarins previously isolated from Macaranga have a methyl substituent (scopoletin)?

Reviewer 3 Report

The manuscript “Chemical Constituents of Macaranga occidentalis, Antimicrobial and Chemophenetic Studies” reports a study in which they evaluate the antimicrobial potential of the crude extract, fractions, and some isolated secondary metabolites from the leaves of Macaranga occidentalis. Seventeen previously known compounds were isolated and exhibited a variety of structural frameworks.  Among these were three steroids, one triterpene, four flavonoids, two stilbenoids, four ellagic acid derivatives, one geraniinic acid derivative, one coumarine, and one glyceride.  The crude extract, fractions, and isolated compounds were all screened for their antimicrobial activity. the crude extract, fractions, and compounds showed varying levels of antibacterial properties against at least one of the tested bacterial strains, with MICs ranging from 250 to 1000 μg/mL.  Among the isolated compounds, schweinfurthin B  exhibited the best activity against Staphylococcus aureus NR 46003, with an MIC value of 62.5 μg/mL.  Schweinfurthin O  and isomacarangin  also exhibited moderate activity against the same 39 strain with a MIC value of 125 μg/mL.

The new compounds were characterized using spectroscopic techniques including FT-IR, 1HNMR, 13C-NMR and HRESIMS.

 General comments:  I was very impressed with the English language and style which is very comprehensible and easy to follow.  The quality and design of figures convey the required information in a very efficient manner.   For instance, Fig 1, 2 and Table 1,2 are quite impressive. I appreciate those figures and tables very much. The results of bioactivity investigations clearly support the conclusions drawn.  The bioactivity results were amply discussed and illustrated in a very nice fashion in Table 2.  All conclusions drawn (related to bioactivity) by the authors are supported by the data and all required results supporting the findings of the paper have been included in the manuscript.  I believe that all experimental protocols/procedure and statistical analysis performed to a high technical standard which are both methodologically and scientifically sound. Characterization of new compounds also meets the criteria outlined by the journal.  The research carried out by the authors will appeal broadly to wide audience in areas such as natural product chemistry, medicinal chemistry, organic chemistry, molecular biology, drug design and drug delivery.  The research is original and has not been published elsewhere. The present work for sure contributes to the  advancement of scientific knowledge. The manuscript is thorough, complete, and will prove a very useful work to share with the scientific community.  The paper is well-organized and the narrative constructive.  One further note is that the manuscript is impressively well written and edited.  Good job for all authors. The manuscript is well suited for molecules. I recommend publication after some revisions especially in relation to structure proof of the reported compounds.

Kindly note the following minor issues:

1.      Line 21 and 22,the affiliation is awkward and confusing.  The email has been repeated twice.  The same email has already been listed in line 7.

2.      Line 64-65. Confusing sentence.  Please re-write.  It seems that the two sentences have been merged together without a period.“treat stomach wash for pregnant women Previous phar-64 macological studies of the crude extracts, fractions and isolated”

3.      In the Characterization of compounds 6a−c section, wouldn’t an AAʹBBʹ system have four doublets? It would be more appropriate based on the NMR to call it an AB system.

4.      In Scheme 2, the allylation mechanism shows an arrow from the alkoxide attacking an sp2 carbon rather than the primary alkyl halide carbon.  What evidence is there to support an SN2’ mechanism?

5.      The authors use C5D5 as NMR solvent?? Do they mean C6D6 ?

6.      The authors use acetone as NMR solvent?? Do they mean deuterated acetone ? please correct

Kindly note the following addition to improve the impact of the manuscript:

The authors have done an incredible job isolating pure compounds as indicated by the provided NMR spectra in the SI section.  The effort is appreciated.  To make the paper useful to organic chemists interested in structures (especially that the bioactivity is not that interesting since it is very low), the authors really need to expand the characterization discussion to include all 17 compounds, not just 6.  The structures of the 17 compounds are complicated and many contain chiral centers.  How was stereochemistry and enantiopurity confirmed? I do not see any optical rotation measurements.   Therefore, the authors should discuss each structure and select and discuss characteristic/distinctive and key signals supporting each structure.  For those with chiral centers, NOESY experiments are required.  Also, the authors should ensure that they have matched their NMR data of all 17 compounds with those published in literature in the same solvent.

Round 2

Reviewer 2 Report

The authors reviewed the requested points and coherently answered the questions addressed. I recommend accepting the manuscript for publication.